# Antidepressant Activities of Synthesized Benzodiazepine Analogues in Mice

**DOI:** 10.3390/brainsci13030523

**Published:** 2023-03-21

**Authors:** Faizan Ul Haq, Mohammad Shoaib, Syed Wadood Ali Shah, Haya Hussain, Muhammad Zahoor, Riaz Ullah, Ahmed Bari, Amal Alotaibi, Muhammad Faisal Hayat

**Affiliations:** 1Department of Pharmacy, University of Malakand, Chakdara, Lower Dir 18800, Khyber Pakhtunkhwa, Pakistan; 2Department of Pharmacy, Shaheed Benazir Bhutto University, Sheringal 18000, Khyber Pakhtunkhwa, Pakistan; 3Department of Biochemistry, University of Malakand, Chakdara, Lower Dir 18800, Khyber Pakhtunkhwa, Pakistan; 4Medicinal Aromatic and Poisonous Plants Research Center, Department of Pharmacognosy, College of Pharmacy, King Saud University, Riyadh 11451, Saudi Arabia; 5Department of Pharmaceutical Chemistry, College of Pharmacy, King Saud University, Riyadh 11451, Saudi Arabia; 6Department of Basic Science, College of Medicine, Princess Nourah bint Abdulrahman University, Riyadh 11671, Saudi Arabia; 7North West Institute of Health and Sciences, Peshawar 25100, Khyber Pakhtunkhwa, Pakistan

**Keywords:** depression, hydroxychalcones, benzodiazepines, antidepressants, forced swim test, tail suspension test, GABAergic system

## Abstract

Depression is a serious psychological disorder which negatively affects human feelings and actions. The use of antidepressants is the therapy of choice while treating depression. However, such drugs are associated with severe side effects. There is a need for efficient and harmless drugs. In this connection, the present study was designed to synthesize several substituted benzodiazepine derivatives and explore their antidepressant potentials in an animal model. The chalcone backbone was initially synthesized, which was then converted into several substituted benzodiazepine derivatives designated as **1**–**6**. The synthesized compounds were identified using spectroscopic techniques. The experimental animals (mice) after acclimatation were subjected to forced swim test (FST) and tail suspension test (TST) after oral administration of the synthesized compounds to evaluate their antidepressant potentials. At the completion of the mentioned test, the animals were sacrificed to determine GABA level in their brain hippocampus. The chloro-substituent compound (**2**) significantly reduced the immobility time (80.81 ± 1.14 s; *p* < 0.001 at 1.25 mg/kg body weight and 75.68 ± 3.73 s with *p* < 0.001 at 2.5 mg/kg body weight dose), whereas nitro-substituent compound (**5**) reduced the immobility time to 118.95 ± 1.31 and 106.69 ± 3.62 s (*p* < 0.001), respectively, at the tested doses (FST). For control groups, the recorded immobility time recorded was 177.24 ± 1.82 s. The standard drug diazepam significantly reduced immobility time to 70.13 ± 4.12 s while imipramine reduced it to 65.45 ± 2.81 s (*p* < 0.001). Similarly, in the TST, the compound **2** reduced immobility time to 74.93 ± 1.14 s (*p* < 0.001) and 70.38 ± 1.43 s (*p* < 0.001), while compound **5** reduced it to 88.23 ± 1.89 s (*p* < 0.001) and 91.31 ± 1.73 s (*p* < 0.001) at the tested doses, respectively, as compared to the control group immobility time (166.13 ± 2.18 s). The compounds **1**, **3**, **4**, and **6** showed weak antidepressant responses as compared to compounds **2** and **5**. The compounds **2** and **5** also significantly enhanced the GABA level in the brain’s hippocampus of experimental animals, indicating the possible involvement of GABAergic mechanism in alleviating the depression which is evident from the significant increase in mRNA levels for the α subunit of the GABA_A_ receptors in the prefrontal cortex of mice as well. From the results, it can be concluded that compound **2** and **5** could be used as alternative drugs of depression. However, further exploration in this connection is needed in other animal models in order to confirm the observed results in this study.

## 1. Introduction

Depression is a psychological disorder that affects human feelings and actions [1]. It can be characterized by anhedonia, distressed sleeping patterns, psychomotor retardation, emotional distress, lack of motivation, decreased social activities, and superiority of oneself with augmented suicidal tendencies [2]. Presently, millions of peoples are suffering from depression around the globe. It is also considered as the main causative factor of suicides committed around the globe [3,4]. Despite its broader distribution worldwide, the accurate etiology of depression is yet to be explored. However, various factors, such as genetic, psychological, and biological influences, may play a vital role in the commencement and preservation of this disorder. The environmental factors, such as exposure to atmospheric chemical pollutants, may increase the probability of incidences of depression in human [5]. Apart from the environmental factors, metabolic, genetic, endocrine, and neurobiological factors are equally important in the etiology of depression. Depression may also leads to a number of serious health complications like diabetes, stroke, Parkinson’s disease, Alzheimer’s disease, multiple sclerosis, obesity, and unexpected cardiac arrest [6,7,8].

Depression develops through a complex mechanism involving many neurological pathways. Out of the several pathways involved in the onset of depression, the dysfunction of glutamatergic and monoaminergic neurotransmission, chronic inflammation, oxidative stress, abnormal synaptic plasticity, and hypothalamic–pituitary–adrenal (HPA) axis are the most important [9].

A number of antidepressant drugs are available, such as selective serotonin reuptake inhibitors (SSRIs), monoamine oxidase inhibitors (MAOIs), tricyclic antidepressants (TCAs), norepinephrine reuptake inhibitors (NRIs), and serotonin and noradrenaline reuptake inhibitors (SNRIs) [10]. These drugs have differential efficacies and are associated with a number of side effects. Plants, being living things, are equipped with natural synthetic machineries which, in disease or unfavorable conditions, produce secondary metabolites. These metabolites enable them to cope with uneven diseased conditions. That is the reason as to why plants have been explored by humans from the beginning of their lives on earth for therapeutic purposes. With the development of modern instruments, the structural elucidation and isolation of the secondary plant metabolites has become possible, which are now even prepared synthetically on larger scales, as in plants they are present in minor quantities. They can be converted into more efficient drugs through synthetic derivatization. As antidepressants, the compounds of natural origin are more efficient if given singly or in combination with existing drugs [11,12]. 

Benzodiazepines (nitrogen containing heterocyclic compounds) are used to treat anxiety, seizures, and insomnia [13,14]. They are also used as analgesic, anticonvulsant, sedative, anti-depressant, and as hypnotic agents [15]. Benzodiazepines like diazepam, clonazepam, chlordiazepoxide, flumazenil, alprazolam, etc., are well-established therapeutic classes of diazepines commonly prescribed to alleviate the core symptoms of anxiety and depression [16]. As antidepressants, these compounds have not been fully evaluated yet.

Keeping in view the aforementioned significances of benzodiazepines in biological system, the current study was conducted to synthesize benzodiazepine derivatives and to investigate them for antidepressant potential as an alternative therapeutic drug of the mentioned psychological disorder. The research work presented herein is novel, as the compound tested for the mentioned potential have not been reported in the literature before.

## 2. Materials and Methods

### 2.1. Materials

All the chemicals and solvents used in this research work were of analytical grade and obtained from Sigma Aldrich, Germany. The reaction progress was monitored by TLC (60F_254_ silica gel; Merck, Germany). The ^1^H-NMR spectra (300 MHz) of the synthesized compounds (in CDCl_3_) were recorded by Bruker Varian Mercury 300 MHz FT Spectrometer (USA). The melting point of the synthesized compounds were determined using Electrothermal 9100 apparatus (Barnstead, UK).

### 2.2. General Procedure of Benzodiazepines Derivatives Preparation

In first step, the synthesis of hydroxychalcones was accomplished by treating various substituted aldehydes with 2-hydroxyacetophenone in cold ethanol in the presence of 40% NaOH with constant stirring. The reaction mixture was neutralized with HCl whereas the reaction progress was monitored by TLC. The filtrate were dried in open air and the dry filtrates were subjected to recrystallization in ethanol [17]. In the second step, various benzodiazepine derivatives were synthesized as per the reported protocol [18,19]. Briefly, the synthesized hydroxychalcones were cyclized with o-phenylenediamine in the presence of triethylamine under proper reaction conditions to synthesize respective benzodiazepine derivatives, as shown in Figure 1.

#### 2.2.1. Synthesis of 2-(4-Phenyl-4,5-dihydro-3H-benzo[b] [1,4] diazepine-2-yl) Phenol (**1**)

The hydroxychalcone backbone of compound **1** was synthesized by mixing 2-hydroxyacetophenone and benzaldehyde in equimolar ratio, followed by its cyclization with o-phenylenediamine in presence of triethylamine. Yield; 72.09%, appearance: light yellow crystalline solid, solubility: ethanol, chloroform, Rf value: 0.54, melting point: 111–114 °C. 1H NMR (400 MHz, CDCl3): δ 15.28 (s, OH), 7.51–7.52 (m, 2H), 7.15–7.41 (m, 6H), 7.21 (td, J = 1.5 Hz, J = 8.3 Hz, 1H), 6.78–7.23 (m, 2H), 6.67 (dd, J = 1.3 Hz, J = 7.9 Hz, 1H), 6.92–6.79 (m, 1H), 5.32 (dd, J = 3.1 Hz, J = 8.0 Hz, 1H), 3.81 (bs, NH), 3.33 (dd, J = 3.7 Hz, J = 13.7 Hz,1H), 3.06 (dd, J = 8.7 Hz, J = 13.6 Hz, 1H) [20].

#### 2.2.2. Synthesis of 4-(o-Hydroxyphenyl)-2-(4-chlorophenyl)-2,3-dihydro-1H-1,5-benzodiazepine (**2**)

The hydroxychalcone backbone of compound **2** was synthesized by mixing 2-hydroxyacetophenone and 4-chlorobenzaldehyde in equimolar ratio, followed by its cyclization with o-phenylenediamine in the presence of triethylamine. Yield; 77.44%, appearance: yellowish crystalline solid, solubility: ethanol, chloroform, Rf value: 0.56, melting point: 171–174 °C. 1H NMR (CDCl3, 400 MHz) δ: 2.97 (dd, 1H, HA, 2 JAM = 16 Hz, 3 JAX = 8 Hz); 3.32 (dd, 1H, HM; 2 JAM = 16 Hz, 3 JMX = 4 Hz); 3.69 (s, 1H, NH); 5.15 (dd, 1H, HX, 3 JAX = 8 Hz, 3 JMX = 4 Hz); 6.88–7.32 (m, 12H, aromatic protons) [21].

#### 2.2.3. Synthesis of 4-(o-Hydroxyphenyl)-2-(4-methoxyphenyl)-2,3-dihydro-1H-1,5-benzodiazepine (**3**)

The hydroxychalcone backbone of compound **3** was synthesized by mixing 2-hydroxyacetophenone and 4-methoxybenzaldehyde in equimolar ratio, followed by its cyclization with o-phenylenediamine in presence of triethylamine. Yield; 71.28%, appearance: yellowish brown crystalline solid, solubility: ethanol, chloroform, Rf value: 0.5, melting point: 133–136 °C. 1H NMR (CDCl3, 400 MHz) δ: 2.97 (dd, 1H, HA, 2 JAM = 16 Hz, 3 JAX = 8 Hz); 3.25 (dd, 1H, HM; 2 JAM = 16 Hz, 3 JMX = 4 Hz); 3.75 (s, 3H, OCH3); 3.76 (s, 1H, NH); 5.08 (dd, 1H, HX, 3 JAX = 8Hz, 3 JMX = 4 Hz); 6.69–7.29 (m, 12H, aromatic protons) [21].

#### 2.2.4. Synthesis of 4-(o-Hydroxyphenyl)-2-(4-toluyl)-2,3-dihydro-1H-1,5-benzodiazepine (**4**)

The hydroxychalcone backbone of compound **4** was synthesized by mixing 2-hydroxyacetophenone and 4-methylbenzaldehyde in equimolar ratio, followed by its cyclization with o-phenylenediamine in presence of triethylamine. Yield; 72.03%, appearance: yellowish orange crystalline solid, solubility: ethanol, chloroform, Rf value: 0.5, melting point: 121–125 ° C.1H NMR (CDCl3, 400 MHz) δ: 2.40 (s, 3H, CH3); 3.19 (dd, 1H, HA, 2 JAM = 8 Hz, 3 JAX = 4 Hz); 3.41 (dd, 1H, HM; 2 JAM = 8 Hz, 3 JMX = 4 Hz); 3.91 (s, 1H, NH); 5.15 (dd, 1H, HX, 3 JAX = 4 Hz, 3 JMX = 4 Hz); 6.81–7.42 (m, 12H, aromatic protons) [21].

#### 2.2.5. Synthesis of 2-[2-(4-Nitrophenyl)-2,3-dihydro-1H-1,5-benzodiazepin-4yl]phenol (**5**)

The hydroxychalcone backbone of compound **5** was synthesized by mixing 2-hydroxyacetophenone and 4-nitrobenzaldehyde in equimolar ratio, followed by its cyclization with o-phenylenediamine in the presence of triethylamine. Yield; 70.12%, appearance: brownish yellow crystalline solid, solubility: ethanol, chloroform, Rf value: 0.53, melting point: 123–126 °C. 1H NMR (δ, DMSO-d6): 6.84–8.31 (m, 12H, Ar-H), 7.79 (s, 1H, -CH=), 8.06 (s, 1H, -NH), 10.91 (s, 1H, -OH) [19].

#### 2.2.6. Synthesis of 2-(4-(3-(Dimethyl amino)phenyl-4,5-dihydro-benzo(1,5)diazepin-2-yl)phenol (**6**)

The hydroxychalcone backbone of compound **6** was synthesized by mixing 2-hydroxyacetophenone and N, *N*-dimethyl benzaldehyde in an equimolar ratio, followed by its cyclization with o-phenylenediamine in the presence of triethylamine. Yield; 73.45%, appearance: reddish-brown crystalline solid, solubility: ethanol, chloroform, Rf value: 0.52, melting point: 136–138 °C. 1H NMR (δ, DMSO-D6): 13.24 s, 1H(OH), 7.9 (s, 1H-ar-H) 7.3–7.8 (m, 3H-ar-H), 6.9–7.3 (m, 2H-ar-H), 6.6–6.9 (m, 6H-ar-H), 4.8–5 (s, 1H=NH), 3.3–3.7 (s, 1H non aromatic,), 2.06 s (2H non aromatic), 1.77 s (6H (CH_3_)2 [22].

### 2.3. In Vivo Pharmacological Activities

#### 2.3.1. Animals

BALB/c albino mice weighing 20–25 g (both sexes) were purchased from NIH (National Institute of Health), Islamabad, and were subjected to acclimatization for 24 h in animal house (free access to water and food, relative humidity = 45–55% and temperature = 20–25 °C). This experimental work was conducted in compliance with the provision of Animal Bylaws 2008 of the University of Malakand, Scientific Procedure Issue-I of Departmental Ethical Committee vide notification no: Pharm/EC/75-01/20.

#### 2.3.2. Acute Toxicity Study of Synthesized Compounds

Acute Toxicity Study was conducted using BALB/c mice of both sexes divided in three groups. In each group there were three animals. Following the OECD guidelines, the animals were given three doses (12.5, 25, and 50 mg/kg body weight) of the synthesized compounds **1**–**6**. The animals were observed for acute toxicity for 72 h (for mortality and morbidity). Out of the tested doses, the 25 mg/kg b.w dose was found safe. Thus, in a subsequent study, a 2.5 mg/kg b.w (which was 1/10th of the safe dose) dose was orally given to animals (divided in six groups) [1,23].

#### 2.3.3. Forced Swim Test

This test was used to evaluate antidepressant effects of the synthesized compounds. The apparatus used in the test is consist of four glass cylinders of 20 cm height and 10 cm diameter. These cylinders are positioned in the center of another white acrylic walled (20 × 40 × 60 cm) squared apparatus. As per the reported procedure, the internal cylinder was filled with hot water up to 7.5 cm [24,25]. The mice were placed gently in this cylinder and the video was recorded for 5 min. The immobility time of each mouse was recorded in seconds. The immobility of mice can be defined as a lack of all types of movements for ≥1 s and floating passively on the surface of water.

#### 2.3.4. Tail Suspension Test

This test was also used to investigate the antidepressant behavior of mice using the previously described protocols [26,27]. The apparatus used was composed of a white acrylic walled (20 × 40 × 60 cm) chamber. The two hanging compartments were separated with an opaque partition at the center of the testing apparatus, in which two mice were suspended simultaneously. Each tested mouse was hung from the tail with adhesive tape < 1 cm from the tip of the tail. The immobility time in a percentage was recorded in seconds for 5 min. The ceasing of all types of struggles for ≥1 s was considered as an immobile phase.

### 2.4. Estimation of Brain GABA Level

The animals were sacrificed immediately after the completion of behavioral experiments. The brain tissues (hippocampus) were isolated and homogenized in 5 mL HCl (0.01 M). The homogenates were maintained at 0 °C for 1 h in 8 mL cold alcohol and then centrifuged for 10 min at 18,000 rpm. The upper phase (3 mL) containing GABA was collected, and 10 µL of it was spotted on Whatman paper. After drying, the spot was eluted with mobile phase containing 50 mL *n*-butanol, 12 mL acetic acid, and 60 mL water following the reported procedure [28]. An ascending technique was used to obtain the chromatogram. Ninhydrin solution (0.5% in 95% ethanol) was applied to visualize the spot. To quantify the GABA components, the blue colored spot was cut and heated for 5 min in water bath. About 5 mL of water was mixed with the spot solution and kept for 1 h. After centrifugation, 2 mL supernatant from mixture was taken and the absorbance was recorded at 570 nm, whereas the GABA contents were estimated from a standard calibration curve [28]. 

### 2.5. Quantitative Real-Time PCR

To further confirm the involved of GABAergic involvement in the under-study process, RNA from the prefrontal cortex of sacrificed mice were isolated. The brain tissues were preserved using RNAlater and stored at a lower temperature for a short period of time. Total RNA was isolated from the tissues using RNeasy Lipid Tissue Mini Kits and its quantification was done spectrophotometrically. Quantitative real-time PCR (qRT-PCR) was used to analyze the gene expression of GABA_A_ receptor (alpha and beta subunits). The primers were designed utilizing the reported procedure [29]. Around 50 µL of reaction mixture containing 25 µL Power SYBRH Green PCR Master Mix kit (Applied Biosystems, Foster City, CA, USA) was subjected to PCR analysis. The specificity of amplification was also confirmed through melting curve analysis. The fold changes of gene expression between the treated and control group were calculated using the ^2-Delta Delta Ct^ methodology [30].

## 3. Results

### 3.1. Synthesis and Characterization of Benzodiazepine Derivatives

As mentioned in the experimental section, initially, the hydroxychalcone backbone of each compound was synthesized by treating various substituted aldehydes with 2-hydroxyacetophenone in cold ethanol in the presence of 40% NaOH (with continuous shaking). TLC was used to monitor the progress of the reaction. The reaction mixtures were then neutralized with 50% HCl solution. The dry filtrates were finally crystallized in ethanol or ethyl acetate. Different substituted benzodiazepine derivatives were then synthesized from the chalcone backbone in the next step. Figure 2 summarizes the chemical structures of compounds **1** to **6**.

The physical parameters of the synthesized benzodiazepine derivatives, such as appearance, solubility, melting point, and percent yield, are summarized in Table 1. The compounds were produced in appropriate amounts and were appreciably soluble in chloroform and ethanol. 

### 3.2. Pharmacological Activities

#### 3.2.1. Acute Toxicity

The synthesized benzodiazepine derivatives (**1**–**6**) were tested for their safety profile in mice. Out of the tested doses, 25 mg/kg body weight was found safe, as no mortality was recoded with this dose. Therefore, in subsequent in vivo experiments, 1.25 and 2.5 mg/kg doses were selected. The selection has been made following OCED guidelines, as described in the experimental section. 

#### 3.2.2. Forced Swim Test

The results of this are summarized in Figure 3. The antidepressant responses were recorded from the observed decrease in the immobility time of the experimental animal in comparison to control group. It was noted that the immobility time was reduced significantly by standard drugs diazepam to 70.13 ± 4.12 s (*p* < 0.001) and imipramine 65.45 ± 2.81 s (*p* < 0.001) being standard drugs. Among the tested compounds, **2** and **5** showed maximum antidepressant activity, which was evident from the reduction in mobility times of the respective group animals (*p* < 0.001) in comparison to the control group at both the tested doses. Compound **4** has also shown antidepressant activity (*p* < 0.01) at 1.25 mg/kg and (*p* < 0.001) 2.5 mg/kg doses. Compound **3** was also active up to some extent and reduced immobility time to 156.51 ± 4.13 s (*p* < 0.01) at 2.5 mg/kg b.w dose. The response of other synthesized compounds was not prominent. Comparatively, compound **2** was more efficient as compared to compound **5**.

#### 3.2.3. Tail Suspension Test

The antidepressant potential of the synthesized benzodiazepine derivatives (**1**–**6**) were also assessed by TST min mice. The results are presented in Figure 4. The standard compounds diazepam and imipramine significantly reduced the mobility times (as compared to control group; 166.13 ± 2.18 s) to 75.93 ± 3.81 s (*p* < 0.001) and 55.43 ± 2.13 s (*p* < 0.001). The synthesized compounds **2** and **5** were found to be promising antidepressant agents, as they significantly reduced (*p* < 0.001) the immobility time of mice at both the tested doses. Compound **1** also showed antidepressant responses (*p* < 0.01) at 1.25 mg/kg and (*p* < 0.001) at 2.5 mg/kg body weight doses. Among the other compounds, **4** and **6** showed moderate antidepressant responses (*p* < 0.05) at 1.25 mg/kg and (*p* < 0.01) at 2.5 mg/kg body weight doses, respectively.

### 3.3. Biochemical Estimation of GABA Level

The synthesized benzodiazepines **2** and **5** significantly reduced the mobility time as predicted by FST and TST, therefore, the animals in the respective groups were sacrificed and brain hippocampus tissues were subjected to the estimation of GABA level after homogenization using an ex vivo approach. A significant increase in GABA level was also recorded for compound **2** (*p* < 0.001) and **5** (*p* < 0.01) at 2.5 mg/kg body weight doses. The standard compound diazepam at 1 mg/kg (*p* < 0.01) and imipramine 60 mg/kg (*p*< 0.001), being the standard drugs, profoundly increased the brain GABA level, as shown in Figure 5A (after FST execution). Similarly, after the TST test, the GABA level was significantly increased by compound **2** (*p* < 0.01) and **5** at the tested doses. Both the standard drugs diazepam and imipramine also significantly (*p* < 0.001) increased the GABA level. 

### 3.4. Quantitative Real-Time PCR

The messenger RNA was obtained from brain tissue of mice and amplified via Quantitative real-time RT-PCR (for groups administered with solvent (control), diazepam, imipramine, compound **2** and compound **5** in 1.25 and 2.5 mg/kg body weight doses). PCR results have been elaborated in Figure 6. The graph demonstrated that compounds **2** and **5** had the highest mRNA expression and binding with α subunits, as opposed to β subunits.

## 4. Discussion

Depression is considered to be the most familiar form of psychological abnormality that is coupled with many manifestations. Depression influences the life standard of individual [31], and can lead to severe stressful conditions [32]. Facial expressions are amongst the contextual factors that can assist in understanding the mental disorders in humans [33]. As mentioned earlier, psychological, biological, and genetic factors are involved in the initiation of depression in humans [5]. Several drugs are used to manage depression and anxiety, administered singly or in combinations. They are mostly associated with adverse side effects and toxicities [34]. As an attempt to design efficient drug for the management of depression, the present study was designed to synthesize the benzodiazepines on the basic nucleus of chalcone. The compounds were obtained in appreciable amounts.

Various animal models have been used for investigating the antidepressant activities of natural and synthetic compounds, utilizing approaches like FST, TST, and Olfactory bulbectomy (OBX), whereas for assessing behavior in anxiety light and dark box, an elevated plus-maze and hole board tests are frequently used [35]. OBX has been reported as a comprehensive model of depression with adequate face and predictive validity and is used to investigate the antidepressant activities precisely [36]. The hole-board assay is a simple procedure for evaluating the reaction of an animal to a strange environment and is extensively used to evaluate emotions in anxiety [37].

The forced swim test and tail suspension test are well-recognized and extensively used while assessing depression in animal models. Both the tests are highly sensitive to all approved classes of antidepressants drugs such as tricyclic antidepressants, selective serotonin reuptake inhibitors, and monoamine oxidases. These tests are commonly used as screening tools for antidepressants in both mice and rats [38]. Moreover, these have been extensively used to investigate the antidepressant effects of both natural and synthetic compounds. As revealed by numerous studies, the FST and TST are reliable in animal models in assessing antidepressant effects of different types of compounds [39,40], therefore these approaches were also applied in assessing the antidepressant potential of compounds **1** to **6**. In recently reported studies, the suspected pharmacological substances were investigated for their antidepressant potentials following FST and TST approaches in animals where significant reduction has been recorded in their immobility times [31,41]. The tests were executed in a stressful environment in which the animal was kept in a stressful condition and become immobile after initial struggling as encountered in a similar fashion in human in case suffering from depression. Among the tested compounds, **2** and **5** with chloro and nitro substituents have shown significant antidepressant potentials as indicated by a decrease in the immobility time estimated through FST and TST.

According to structure activities relationships studies, any substitution on rings with electron withdrawing groups like halogen or nitro groups enhances the biological activities whereas the electron donating groups like methyl and methoxy groups causes reduction in the biological activities of benzene ring containing compounds. In a similar fashion, compounds **2** and **5** have shown enhanced antidepressant activities due to the presence of electron-withdrawing groups attached to the ring [42,43].

Experimental evidence has suggested that the GABAergic system is involved in the pathophysiology of depression and anxiety. Moreover, it has been reported that depressed patients have diminished GABA receptors or GABA levels. Thus, substances that may have positive allosteric effects on GABAergic system may be useful in the treatment of depression [44]. The GABAergic system is constituted of γ-Aminobutyric acid (GABA), an inhibitory neurotransmitter of the central nervous system [45]. The reported studies have revealed that nitrogen-containing heterocyclic compounds like isoxazole, pyrazole, pyrimidine, diazepines, carbazole, and oxazepanes, etc., important constituents of a wide variety of natural products with diverse pharmacodynamic applications, are excellent antidepressant agents [46]. Our compounds being heterocyclic in a similar fashion have produced antidepressant effects. The GABAergic mechanism involvement was further confirmed through PCR, where promising results were shown by compounds **2** and **5**, exhibiting significantly high mRNA expression and binding with α subunits as compared to β subunit. Our finding were in agreement with a previously reported study [47].

Although the present research work has exhibited the antidepressant properties of the synthesized benzodiazepine derivatives, these are preliminary findings and are only limited to the mice model with limited utilization of advanced scientific techniques. Further studies are needed to explore the exact mechanism in other animal models, whereas the use of sophisticated advanced research tools is equally important. 

## 5. Conclusions

In this study, the synthesized benzodiazepine derivatives have relieved the symptoms of depression and stress in mice model, especially by compounds **2** and **5**. The antidepressant potential of these compounds was most probably due to the enhanced GABA level in brains as observed, suggesting the possible involvement of GABAergic mechanism. The claims made at this stage are quite preliminary and need to be confirmed in another animal model as well. Additionally, other biological potentials of these compounds need to be investigated as well.

## Figures and Tables

**Figure 1 brainsci-13-00523-f001:**
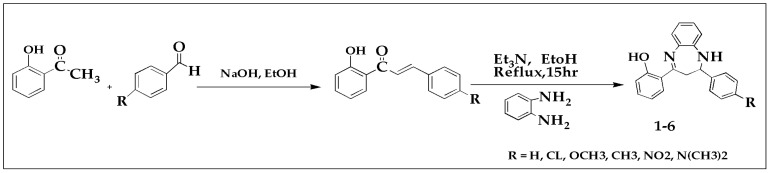
Synthesis scheme of benzodiazepines derivatives (**1**–**6**).

**Figure 2 brainsci-13-00523-f002:**
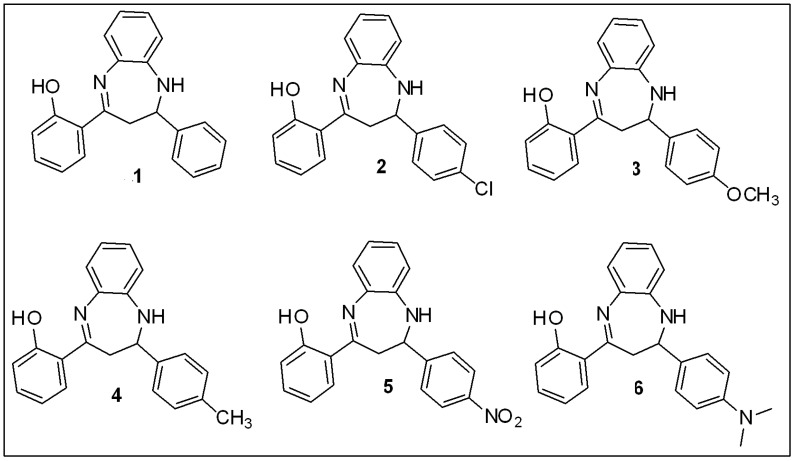
Chemical structures of the synthesized benzodiazepine derivatives (**1**–**6**).

**Figure 3 brainsci-13-00523-f003:**
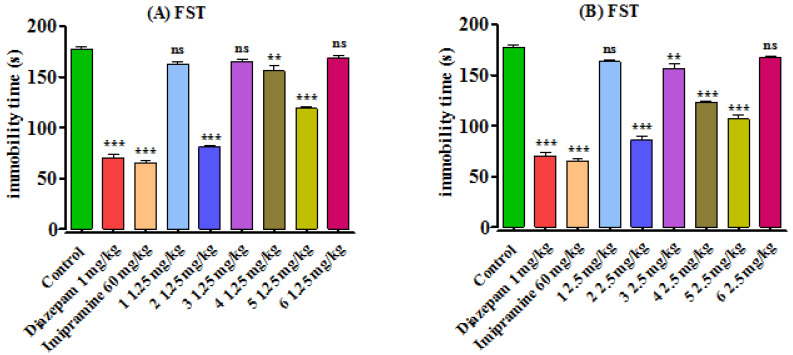
Effects of synthesized benzodiazepines (**1**–**6**) on the immobility time in mice using FST. Results are shown as Mean ± SEM, *n* = 8, while *p* ˃ 0.05 ‘ns, *p* < 0.01 **, and *p* < 0.001 ***. Mice were administered with synthesized benzodiazepine derivatives (**1**–**6**) in 1.25 (**A**) and 2.5 mg/kg (**B**) body weight doses, diazepam in 1 mg/kg, and imipramine 60 mg/kg body weight. One-way ANOVA followed by Bonferroni’s multiple comparison tests was applied to decide the significance level of the data.

**Figure 4 brainsci-13-00523-f004:**
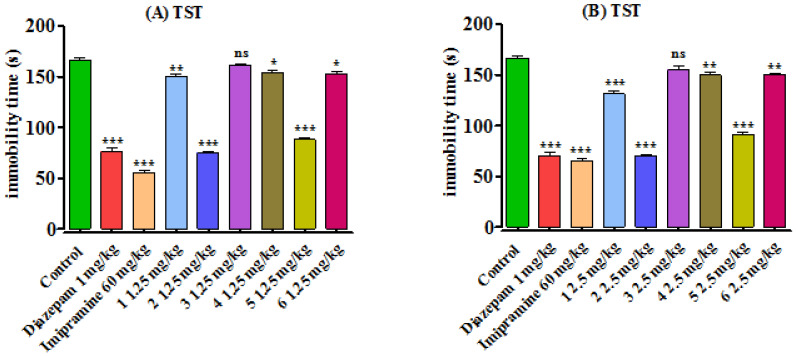
Effects of synthesized benzodiazepines (**1–6**) on the immobility time in mice using TST. Results are shown as Mean ± SEM, *n* = 8, while *p* ˃ 0.05 ‘ns, *p* < 0.05 *, *p* < 0.01 **, and *p* < 0.001 ***. Mice were administered with synthesized benzodiazepine derivatives (**1–6**) in 1.25 (**A**) and 2.5 mg/kg (**B**) body weight doses, diazepam in 1 mg/kg, and imipramine 60 mg/kg body weight. One-way ANOVA, followed by Bonferroni’s multiple comparison tests, was applied to decide the significance level of the data.

**Figure 5 brainsci-13-00523-f005:**
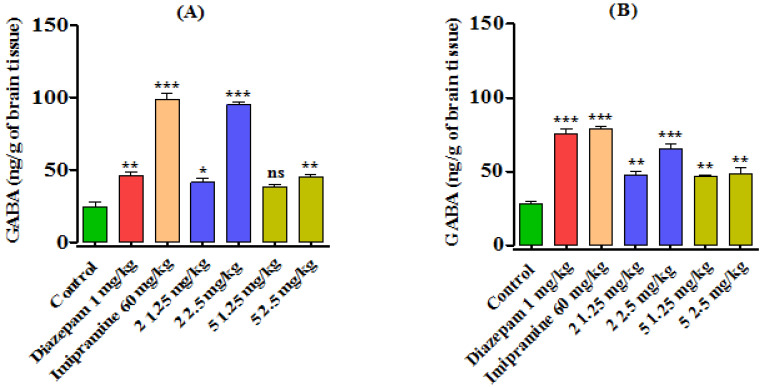
Effects of synthesized benzodiazepines **2** and **5** on brain GABA level (**A**) after (FST) and (**B**) after TST test. The results were shown as Mean ± SEM, *n* = 8. The level of significance (*p* ˃ 0.05 ‘ns, *p* < 0.05 *, *p* < 0.01 **, and *p* < 0.001 ***) vs. control was recorded using One-way ANOVA followed by Bonferroni’s multiple comparison test.

**Figure 6 brainsci-13-00523-f006:**
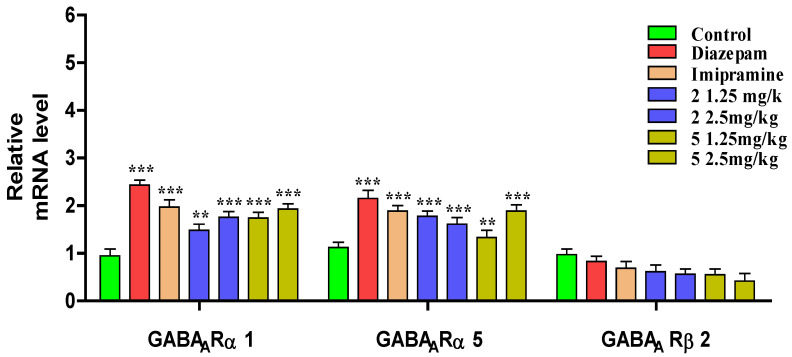
mRNA expression of binding of synthesized benzodiazepines **2** and **5** with GABA with PCR. The results were shown as Mean ± SEM, *n* = 8. The level of significance (*p* < 0.01 **, and *p* < 0.001 ***) vs. control has been recorded using One-way ANOVA, followed by Bonferroni’s multiple comparison test.

**Table 1 brainsci-13-00523-t001:** Physical parameters of the synthesized benzodiazepines derivatives (**1**–**6**).

Sample	Appearance	Rf	Solubility	Melting Point °C	%Yield
**1**	Crystalline, Light-yellow solid	0.54	Ethanol/chloroform	111–114	72.09
**2**	Crystalline, Yellowish Brown solid	0.56	Ethanol/chloroform	171–174	77.44
**3**	Crystalline, Yellowish Brown solid	0.5	Ethanol/chloroform	133–136	71.28
**4**	Crystalline, Yellowish Orange solid	0.5	Ethanol/chloroform	121–125	72.03
**5**	Crystalline, Yellowish Brown solid	0.53	Ethanol/chloroform	123–126	70.12
**6**	Crystalline, Reddish Brown solid	0.52	Ethanol/chloroform	136–138	73.45

## Data Availability

Not applicable.

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
