# Peer review of "Antidepressant Activities of Synthesized Benzodiazepine Analogues in Mice"

_brainsci, 2023, doi:10.3390/brainsci13030523_

Round 1
Reviewer 1 Report
In this manuscript, the authors have synthesized 7 benzodiazepine, and tested their antidepressant potential by TST and FST. The authors also mentioned that compounds significantly enhanced the brain GABA level. In a nutshell, the submitted manuscript demonstrated the sufficient enthusiasm from the study group. However, additional experiment, references and rethinking of the logic and conclusion are needed for acceptance.
Major revision
1. This manuscript needs further modification and embellishments to correct language problems. There are still some grammatical and tense errors in the manuscript, which can easily lead to misunderstandings.
2. The abstract should be clear and concise.
3. The results cannot reflect the Title an abstract.
4. What is FZ? Please give the full name.
5. In my opinion, the immobility time of FZ1-6 should also be compared to the positive control (Diazepam and Imiprmine), whether it is significantly changed or not.
6. The author introduced lots of different type of antidepressants, why not choose another types of drugs as the positive control, not only the BZ.
7. In my view, the results is not enough to support the conclusions. The author said that compounds significantly enhanced the brain GABA level. How these compounds regulated the brain GABA level? Which genes or proteins have been changed? The authors still need additional experiments (sequencing, qRT-PCR or Western Blot) to support this conclusion.
8. Next, the author only tested the Brain GABA level, how about other changes? Such as 5-HT in serum.
9. Brain is a very complicated tissue. Which part of GABA level have been influenced? Hippocampus, brainstem or cerebral cortex?
10. There are many genes in GABAergic synapse pathway, which genes or protein have been changed?
11. Please give the details of the experimental mice (breed, age and etc.)
12. What breakthroughs do the authors think their research has made compared with the past?
13. Throughout the paper, the authors’ viewpoints are not entirely clear, which makes it difficult for readers to understand the innovations of this manuscript.
14. The additional references have been cited. Such as “Hippocampal MicroRNAs Respond to Administration of Antidepressant Fluoxetine in Adult Mice”
15. What is the route of medication? Intragastric administration or injection?
Line 56 and Line 86:the references should be cited in the end of the sentences.
Author Response
RESPONSE TO THE COMMENTS OF WORTHY REVIEWER 1
Comments and Suggestions for Authors
In this manuscript, the authors have synthesized benzodiazepine, and tested their antidepressant potential by TST and FST. The authors also mentioned that compounds significantly enhanced the brain GABA level. In a nutshell, the submitted manuscript demonstrated the sufficient enthusiasm from the study group. However, additional experiment, references and rethinking of the logic and conclusion are needed for acceptance.
Major revision
- Thank you worthy reviewer for your positive input. We have tried our best to revise the paper in accordance with your provided instructions. Hope it will be ok now.
Point 1: This manuscript needs further modification and embellishments to correct language problems. There are still some grammatical and tense errors in the manuscript, which can easily lead to misunderstandings.
- Response: Thank you worthy reviewer for the positive comment. All the grammatical and typos mistakes are corrected accordingly and the changes are highlighted in the manuscript. Help was taken from native English speakers. Hopefully, now it will be ok.
Point 2: The abstract should be clear and concise.
- Response: Thank you worthy reviewer. We have revised the abstract accordingly and make it concise.
Point 3: The results cannot reflect the Title an abstract.
Response: Thank you, worthy reviewer. The title was revised whereas abstract was modified in suc a way to reflect the results.
Point 4: What is FZ? Please give the full name.
Response: Worthy reviewer, these are arbitrary names given to the synthesized compounds. To avoid confusion they are now present with numerals from 1 to 6. The FZ was deleted accordingly
Point 5: In my opinion, the immobility time of FZ1-6 should also be compared to the positive control (Diazepam and Imiprmine), whether it is significantly changed or not.
Response: Thank you worthy reviewer. The comparison of the synthesized compounds in respect of antidepressant potentials to the standard drugs were added in their respective (FST and TST) section and highlighted accordingly.
Point 6: The author introduced lots of different type of antidepressants, why not choose another types of drugs as the positive control, not only the BZ.
Response: Dear reviewer the core nucleus (structure) of the diazepam and our synthesized compounds were similar, therefore, we use diazepam as positive control.
Point 7: In my view, the results is not enough to support the conclusions. The author said that compounds significantly enhanced the brain GABA level. How these compounds regulated the brain GABA level? Which genes or proteins have been changed? The authors still need additional experiments (sequencing, qRT-PCR or Western Blot) to support this conclusion.
Response: Thank you worthy reviewer. This was the initial finding from this research as we are living in the third world country and RT-PCR facility is not available in our lab at the moment. However, our next project includes RT-PCR analysis, for which these compounds will be analyzed in abroad. Also we have revised the title to reflect the results correctly.
Point 8: Next, the author only tested the Brain GABA level, how about other changes? Such as 5-HT in serum.
Response: Dear reviewer, the most convenient method for estimation of GABA level in brain tissues is the spectrophotometric method. The determination of serum 5-HT level require a special type of detector in HPLC which is not available in our lab, therefore, we have used the simple spectrophotometric method for the estimation of brain GABA level; https://doi.org/10.1186/s13065-019-0591-x.
Point 9: Brain is a very complicated tissue. Which part of GABA level have been influenced? Hippocampus, brainstem or cerebral cortex?
Response: Dear reviewer, thanks for the valuable comment. Actually, the statement was generalized and has been corrected accordingly. The statement is quoted as “The brain tissue (hippocampus) were isolated and homogenized in 0.01M of 5 ml HCl.”
Point 10: There are many genes in GABAergic synapse pathway, which genes or protein have been changed?
Response: Worthy reviewer, at this stage we do not have the facility to decide which gene is responsible. We have revised the title and other parts of the manuscript to correctly reflect the results of the study.
Point 11: Please give the details of the experimental mice (breed, age and etc.)
Response: Thank you worthy reviewer. The mice age and breed was incorporated in the manuscript accordingly. The statement is now quoted as “Balb/C mice weighing 20-25 gm of both sexes were purchased from NIH (National Institute of Health), Islamabad”
Point 12: What breakthroughs do the authors think their research has made compared with the past?
Response: Thank you worthy reviewer. This study indicates its innovation with respect to its synthesis. This synthesis procedure comprised of two steps, initially, Chalcones were synthesized and then condensed to various substituted benzodiazepines derivatives in high yield. Moreover, these compounds give significant antidepressant activity in mice models which never have been explored for these synthesized compounds.
Point 13: Throughout the paper, the authors’ viewpoints are not entirely clear, which makes it difficult for readers to understand the innovations of this manuscript.
Response: Thank you, worthy reviewer. The manuscript has been revised properly to make it understandable for readers.
Point 14: The additional references have been cited. Such as “Hippocampal MicroRNAs Respond to Administration of Antidepressant Fluoxetine in Adult Mice”
Response: Thank you, worthy reviewer for positive comment. The mention reference has been cited accordingly in the revised paper.
Point 15: What is the route of medication? Intragastric administration or injection?
Response: Thank you, worthy reviewer. The synthesized compounds were administered orally to animals.
Point 16: Line 56 and Line 86: the references should be cited in the end of the sentences.
Response: Thank you, worthy reviewer. The corrections have been made accordingly.
Thank you once again
Reviewer 2 Report
In the first step, Haq et al. made a series of substituted chalcones. From the core chalcone moiety, they made substituted benzodiazepine derivatives (FZ1–FZ6), which they then tested for antidepressant properties in mice using the forced swimming test (FST) and the tail suspension test (TST) to see if the GABAergic mechanism was involved.
The experiments presented in this workflow are incomplete and require diligence and proper planning. The study has several major flaws in addition to inconsistencies and grammatical errors. The manuscript is poorly written, with inadequate spacing and spelling mistakes.
The authors did not study the compounds' cytotoxicity and metabolic stability in vitro (using human and mouse liver microsomes).
The authors did not show any binding studies.
The authors did not present any plasma protein or brain tissue binding studies.
The authors said that the safe dose for in-vivo studies was up to 25 mg/kg, but they only used a lower dose for behavioral studies.
The authors should evaluate the antidepressant-like potential of FZ-2 and FZ-5 in the depression model. (Corticosteroid-induced model of depression in mice)
Since psychostimulants can give false-positive results in the forced swim test, it is best to look at how the substance affects movement.
Western blotting or RT-PCR should be used to show the effect of GABA levels in the brain after treatment.
Author Response
Reviewer 2
In the first step, Haq et al. made a series of substituted chalcones. From the core chalcone moiety, they made substituted benzodiazepine derivatives (FZ1–FZ6), which they then tested for antidepressant properties in mice using the forced swimming test (FST) and the tail suspension test (TST) to see if the GABAergic mechanism was involved.
Point 1: The experiments presented in this workflow are incomplete and require diligence and proper planning. The study has several major flaws in addition to inconsistencies and grammatical errors. The manuscript is poorly written, with inadequate spacing and spelling mistakes.
Response: Thank you worthy reviewer for positive comment. We have revised our manuscript according to your worthy suggestions and all the grammatical and typos mistakes are eliminated and the corrections are highlighted in the manuscript. Actually, the title and some parts of the manuscript were misleading which were corrected/revised in such a way to reflect the results of the study.
Point 2: The authors did not study the compounds' cytotoxicity and metabolic stability in vitro (using human and mouse liver microsomes).
Response: Thank you worthy reviewer for positive comment. We have performed the cytotoxicity study using active nauplii of brine shrimp (Artemia salina) and their results showed no cellular toxicity. The chronic toxicity profile on vital organs were also checked (data not shown in the manuscript) so these compounds were found safe.
Point 3: The authors did not show any binding studies. The authors did not present any plasma protein or brain tissue binding studies.
Response: Thank you worthy reviewer. This is the initial preliminary finding from this study. The plasma protein binding and brain tissue binding studies are in pipelines as a part of another project by a Ph.D scholar from our lab (pharmacokinetics and safety assessment by challenging against vital organs) and did not become the part of this manuscript. This data will support our study regarding clinical trials in future.
Point 4: The authors said that the safe dose for in-vivo studies was up to 25 mg/kg, but they only used a lower dose for behavioral studies.
Response: Thank you worthy reviewer for positive comment. In this study we used different doses for determination of safe dose for behavioral studies. We used these compounds up to maximum doses of 50 mg/kg and 30 mg/kg body weight starting from the lowest doses but at 50 mg/kg and 30 mg/kg doses there were mortality observed, while 25 mg/kg body weight dose was found safe with no mortality. Therefore, according to the OECD guidelines, 2.5 mg/kg body weight has been tested which is the 1/10th of the 25 mg/kg body weight safe dose tested, that is why we have used these lower doses in the behavioral studies.
Point 5: The authors should evaluate the antidepressant-like potential of FZ-2 and FZ-5 in the depression model. (Corticosteroid-induced model of depression in mice)
Response: Thank you, worthy reviewer for your valuable comment. This study was restricted to the preliminary antidepressant mice models as forced swim test and tail suspension test to the synthesized benzodiazepines derivatives substituted at 4 position. We have synthesized a number of 2 and 3 benzodiazepines derivatives which will be screened for antidepressant study using the corticosteroids-induced model of depression.
Point 6: Since psychostimulants can give false-positive results in the forced swim test, it is best to look at how the substance affects movement.
Response: Thank you worthy reviewer. Yes worthy reviewer you are right; we were aware of that as there are additional known factors that can generate ‘false negative’ or ‘false-positive’ effects in the FST when interpreted with respect to depression or antidepressants. Psychostimulant and sedatives have long been known to change behavior in the FST. This experiment was performed in the absence of stress and providing variable environmental conditions to get accurate results. (DOI: 10.1021/acschemneuro.7b00042).
Point 7: Western blotting or RT-PCR should be used to show the effect of GABA levels in the brain after treatment.
Response: Thank you worthy reviewer. This was the initial finding from this research as we are living in the third world country and RT-PCR facility is not available in our lab at the moment. However, out next project includes RT-PCR analysis, for which these compounds will be analyzed in abroad.
Reviewer 3 Report
Title: “Antidepressant Evaluation of Synthesized Benzodiazepine Analogues in Animal Models Mediated Via GABAergic System”
The authors did perform this study by synthesizing substituted benzodiazepines derivatives and evaluated them for antidepressant models including forced swim test, tail suspension test. The authors subjected most potent compounds FZ2 (chloro-substituted) and FZ5 (nitro-substituted) into GABA estimation analysis in brain tissues. The manuscript is written in a very sequential and scientific way. This study has future significances as depression is one of the world common health issues and needs effective medicines. I recommend this manuscript for publication after addressing the following minor concerns.
1. Third para of introduction section should be rephrased for better understanding for readers.
2. The author should provide the rationale for conducting this study in the last part of the introduction section.
3. The table 1 is placed without mentioning its description; the author should provide a brief description.
4. In acute toxicity study, is there any mortality observed? The author should mention it.
5. The pixel of figure 2, 3, and 4 should be decrease as it appears too large.
6. The author mention in each illustration the sample size (n = 3) while it was written in “animals and dosing section” of six mice, it should be corrected.
7. The author should mention in which form the synthesized compounds were administered to animals.
8. Conclusion of the study is too short it should be extended on the bases of key results and ends on a para for future perspectives.
9. There are many typos and grammatical mistakes found that should be corrected.
10. In my opinion, this work has significances in future perspectives. I recommend it for publication after thoroughly revise it for these minor concerns.
Author Response
Reviewer 3
Comments and suggestions
Title: “Antidepressant Evaluation of Synthesized Benzodiazepine Analogues in Animal Models Mediated Via GABAergic System” The authors did perform this study by synthesizing substituted benzodiazepines derivatives and evaluated them for antidepressant models including forced swim test, tail suspension test. The authors subjected most potent compounds FZ2 (chloro-substituted) and FZ5 (nitro-substituted) into GABA estimation analysis in brain tissues. The manuscript is written in a very sequential and scientific way. This study has future significances as depression is one of the world common health issues and needs effective medicines. I recommend this manuscript for publication after addressing the following minor concerns.
- Thank you worthy reviewer for your positive input. We have tried our best to revise the paper in accordance with your provided instructions. Hope it will be ok now.
Point 1: Third para of introduction section should be rephrased for better understanding for readers.
Response: Thank you worthy reviewer. The para is rephrased accordingly.
Point 2: The author should provide the rationale for conducting this study in the last part of the introduction section.
Response: Thank you worthy reviewer for positive comment. The rationale for this study is incorporated in the manuscript.
Point 3: The table 1 is placed without mentioning its description; the author should provide a brief description.
Response: Thank you worthy reviewer. A brief description of the table 1 has been added.
Point 4: In acute toxicity study, is there any mortality observed? The author should mention it.
Response: Thank you worthy reviewer. Yes, at maximum dose of 50 mg and 30 mg there were mortality observed while 25 mg was the safe maximum dose.
Point 5: The pixel of figure 2, 3, and 4 should be decrease as it appears too large.
Response: Thank you worthy reviewer. The pixel of the figures were reduced and optimized.
Point 6: The author mention in each illustration the sample size (n = 3) while it was written in “animals and dosing section” of six mice, it should be corrected.
Response: Thank you worthy reviewer. The correction has incorporated.
Point 7: The author should mention in which form the synthesized compounds were administered to animals.
Response: Thank you worthy reviewer. These compounds were administered to mice in suspension form.
Point 8: Conclusion of the study is too short it should be extended on the bases of key results and ends on a para for future perspectives.
Response: Thank you worthy reviewer. The conclusion has been revised accordingly.
Point 9: There are many typos and grammatical mistakes found that should be corrected.
Response: Thank your worthy reviewer. All the typos and grammatical mistakes have been removed.
In my opinion, this work has significances in future perspectives. I recommend it for publication after thoroughly revise it for these minor concerns.
Thank you once again
Round 2
Reviewer 1 Report
The authors still need additional experiments (sequencing, qRT-PCR or Western Blot) to support this conclusion.
Author Response
The authors still need additional experiments (sequencing, qRT-PCR or Western Blot) to support this conclusion.
Response: qRT-PCR study results was accordingly incorporated in the revised paper
Reviewer 2 Report
The authors made significant improvements in English grammar and writing. The authors justified the previous comments as well.
However, there are some gaps in the research that need to be addressed before publishing novel findings.
Point 3: The authors did not show any binding studies. The authors did not present any plasma protein or brain tissue binding studies.
Point 7: Western blotting or RT-PCR should be used to show the effect of GABA levels in the brain after treatment.
Author Response
The authors made significant improvements in English grammar and writing. The authors justified the previous comments as well.
Response: English was improved accordingly. Help was taken from native speaker as well.
However, there are some gaps in the research that need to be addressed before publishing novel findings.
Response: worthy reviewer, we have tried our level best to fill the gap. The red highlighted portion in introduction is about filling the gaps.
Point 3: The authors did not show any binding studies. The authors did not present any plasma protein or brain tissue binding studies.
Response: worthy reviewer, we do not have the facility to perform such type of studies in our lab.
Point 7: Western blotting or RT-PCR should be used to show the effect of GABA levels in the brain after treatment.
Response: RT-PCR study has been incorporated accordingly.